# Hydrothermal Stability of Hydrogen-Selective Carbon–Ceramic Membranes Derived from Polybenzoxazine-Modified Silica–Zirconia

**DOI:** 10.3390/membranes13010030

**Published:** 2022-12-26

**Authors:** Sulaiman Oladipo Lawal, Hiroki Nagasawa, Toshinori Tsuru, Masakoto Kanezashi

**Affiliations:** Chemical Engineering Program, Graduate School of Advanced Science and Engineering, Hiroshima University, 1-4-1 Kagamiyama, Higashi-Hiroshima 739-8527, Japan

**Keywords:** benzoxazine, carbon–ceramic, pyrolysis, hydrothermal stability, silica–zirconia, hydrogen

## Abstract

This work investigated the long-term hydrothermal performance of composite carbon-SiO_2_-ZrO_2_ membranes. A carbon-SiO_2_-ZrO_2_ composite was formed from the inert pyrolysis of SiO_2_-ZrO_2_-polybenzoxazine resin. The carbon-SiO_2_-ZrO_2_ composites prepared at 550 and 750 °C had different surface and microstructural properties. A carbon-SiO_2_-ZrO_2_ membrane fabricated at 750 °C exhibited H_2_ selectivity over CO_2_, N_2_, and CH_4_ of 27, 139, and 1026, respectively, that were higher than those of a membrane fabricated at 550 °C (5, 12, and 11, respectively). In addition to maintaining high H_2_ permeance and selectivity, the carbon-SiO_2_-ZrO_2_ membrane fabricated at 750 °C also showed better stability under hydrothermal conditions at steam partial pressures of 90 (30 mol%) and 150 kPa (50 mol%) compared with the membrane fabricated at 500 °C. This was attributed to the complete pyrolytic and ceramic transformation of the microstructure after pyrolysis at 750 °C. This work thus demonstrates the promise of carbon-SiO_2_-ZrO_2_ membranes for H_2_ separation under severe hydrothermal conditions.

## 1. Introduction

Due to energy security and environmental concerns [1,2], hydrogen has been considered as both a clean and renewable source of energy to replace fossil fuels [3,4,5]. Hydrogen possesses a high energy density and the ability to form covalent bonds with other elements (such as nitrogen) for easy storage, transportation, and feedstock for other industrial chemical processes [5,6]. Hydrogen can electrochemically combine with oxygen from air in fuel cells to generate electricity with clean H_2_O effluent [5,7]. Hydrogen can be generated from the chemical conversion of some hydrocarbon compounds. An advantage of this production route is the ability to produce hydrogen on a large scale and subsequently convert it to a transportable form in integrated chemical plants. Equations (1)–(4) show examples of important chemical reactions, which involve dehydrogenation and steam reforming of hydrocarbons.

Dehydrogenation of propane
C_3_H_8_ ↔ C_3_H_6_ + H_2_(1)

Methane steam reforming
CH_4_ + 2H_2_O ↔ CO_2_ + 4H_2_(2)

Ethanol steam reforming
C_2_H_5_OH + 3H_2_O ↔ 2CO_2_ + 6H_2_(3)

Dehydrogenation of methyl cyclohexane
C_6_H_11_CH_3_ ↔ C_6_H_5_CH_3_ + 3H_2_(4)

Separation of hydrogen from other side product species, unreacted reactants, and intermediate reaction species is important. Appendix A presents a comparison of popular H_2_ separation technologies, which include pressure swing adsorption (PSA), cryogenic distillation [8,9,10], and the recently developed inexpensive and energy-efficient membrane-separation processes [9,10]. Membranes for H_2_ separation have been widely reviewed [9,11,12] and membranes with H_2_ permeability of 1000 Barrer and H_2_/X selectivity ≥ 100 are desired [12]. Studies on ceramic membranes derived from SiO_2_ have highlighted their promise in this quest [9,11]. However, maintaining such high performance under hydrothermal conditions has been one of the major problems facing H_2_ separation membranes [13,14,15]. For example, Lin et al. reported that SiO_2_ membranes exposed to 50 mol% of steamed atmosphere at 600 °C for 30 h exhibited losses of 48 and 77% in specific surface area and pore volume, respectively [13]. Lin’s group also reported similar effects on ZrO_2_, TiO_2_ and Al_2_O_3_-derived membranes [14]. These structural changes are deemed to be due to the rearrangement of networks initiated from hydrolysis by water vapor and from a subsequent recondensation of the formed -OH groups [16].

Carbonized ceramic membranes have been proposed as promising candidates that could perform long-term under hydrothermal conditions for hydrogen production. Duke and co-workers found that the carbonization of surfactant-templated silica (hexyl trimethyl ammonium bromide–silica) improved the hydrothermal stability of a silica membrane quite significantly [17,18]. The amorphous carbon nanoparticles that formed inhibited the silanol (-Si-OH) migration and condensation that results in closure of the microporous structure. In addition to this, the presence of the carbon nanoparticles also served to increase the pore volume required for enhanced H_2_ permeability [17,19]. However, the surfactant carbonization route did not allow pore size tuning of the silica structure for targeted applications [17].

Another method for the carbonization of ceramic networks involves the use of organic coordination ligands to form coordinate complexes where the precursor to be modified is a transition metal ion compound [20,21,22]. Molecular sieving membranes were successfully fabricated by using an acetylacetone ligand to modify SiO_2_-ZrO_2_ [23], and the fabrication of hydrogen-selective carbon-SiO_2_-ZrO_2_ membranes was later studied by pyrolysis of the acetylacetone ligand at 550 °C [19,24]. Carbon-SiO_2_-ZrO_2_ membranes have shown promise for high-temperature H_2_/CO_2_ separation performance [24], whereas the lower pyrolysis temperature of acetylacetone and its non-thermosetting properties prevents the formation of molecular sieving that is required in hydrothermal applications at much higher temperatures. Therefore, the fabrication of a hydrothermally stable and molecular sieving carbon-SiO_2_-ZrO_2_ membrane required a better carbon precursor. In a more recent work, we studied the novel use of a traditional thermosetting group of organic compounds known as benzoxazines as the chelating ligand for the modification and carbonization of the SiO_2_-ZrO_2_ network [25]. A carboxylic benzoxazine (3-(3-oxo-1,4-benzoxazin-4-yl) propanoic acid) possessing a hydroxylated nucleophilic carboxyl group was specifically chosen as the chelating ligand. This enabled the formation of a composite SiO_2_-ZrO_2_-polybenzoxazine resin material that was used to fabricate thin carbon-SiO_2_-ZrO_2_ membranes with molecular sieving performances that could be tuned via pyrolysis temperature. H_2_/SF_6_ and H_2_/CH_4_ selectivity ranged from 10^2^ to 10^4^ and from 10 to 10^2^, respectively, depending on pyrolysis temperatures that ranged between 300 and 850 °C [25]. The wide range of the pyrolysis temperature for this SiO_2_-ZrO_2_-polybenzoxazine (SZ-PB) resin presented an opportunity for the fabrication of carbon–ceramic composite membranes at higher than the application temperatures that are usually required for a stable performance.

In this work, we therefore investigated the long-term hydrothermal applications of SiO_2_-ZrO_2_-polybenzoxazine-derived carbon-SiO_2_-ZrO_2_ (C-SZ) membranes. First, we established an improved formation strategy for the preparation of SiO_2_-ZrO_2_-polybenzoxazine preceramic resin. The effects of the pyrolysis temperature of SiO_2_-ZrO_2_-polybenzoxazine at 550 and 750 °C on the resulting C-SZ surface properties and microstructure were studied and supported with various characterizations. Hydrothermal stability experiments were carried out to evaluate membrane properties at specific intervals. H_2_O + N_2_ mixtures (90 and 150 kPa partial pressure of steam) were fed to the membranes at 500 °C for several hours until steady permeance was observed. Membrane permeation properties before and after each steam treatment experiment were then compared to evaluate the progress of hydrothermal stability. Herein we propose mechanisms to explain the observed results.

## 2. Experimental Section

### 2.1. Materials

A precursor resin sol was prepared via a sol–gel process. Vinyltrimethoxysilane (VTMS; JNC Co. Ltd., Tokyo, Japan; 98% purity) was used as the silica precursor while zirconium n-butoxide (ZrTB; Sigma-Aldrich Japan, Tokyo, Japan; 80% in butanol) served as the zirconia precursor. 3-(3-oxo-1,4-benzoxazin-4-yl) propanoic acid (BZPA; Sigma-Aldrich Japan, Tokyo, Japan; 99%) was utilized as a chelating ligand for ZrTB modification. The solvent medium used to carry out the reactions was composed of a 50:50 mixture of dimethyl carbonate (DMC; Nacalai Tesque, Kyoto, Japan) and ethanol (EtOH; Sigma-Aldrich). In the sol–gel reactions, hydrochloric acid (HCl; Nacalai Tesque, Kyoto, Japan; 37% pure) served as the catalyst for hydrolysis. Dibenzoyl peroxide (BzO_2_; Sigma-Aldrich Japan, Tokyo, Japan) was used as a radical initiator during thermal curing. All materials were used as received without further purification.

### 2.2. Preparation of SiO_2_-ZrO_2_-Polybenzoxzine (SZ-PB) Precursor Resin Sol and of Carbon-SiO_2_-ZrO_2_ (C-SZ) Films, Powders, and Membranes

First, 2 wt% of SZ-PB was prepared in two stages. In the first stage, ZrTB dissolved in equal parts DMC and ethanol was modified via a reaction with BZPA (BZPA/ZrTB molar ratio 4:1) for one hour at room temperature. In the second stage, a solution of VTMS in equal parts DMC and ethanol was then co-hydrolyzed with the BZPA-modified ZrTB (Si/Zr molar ratio 9:1) using deionized water (H_2_O/alkoxide molar ratio 4) and HCl as a catalyst (H^+^/alkoxide molar ratio, 1:4). To accomplish hydrolysis and poly-condensation, the mixture was stirred at 600 rpm for more than 12 h at room temperature.

After the hydrolysis and polycondensation reactions, dibenzoyl peroxide (BzO_2_) was added (radical initiator/BZPA molar ratio 0.1) as a curing agent. The resulting sol was thermally cured dropwise in a platinum plate heated to 200 °C to obtain the SZ-PB preceramic resin gel. Carbon-SiO_2_-ZrO_2_ powders were then prepared from the SZ-PB preceramic resin gel via pyrolysis at 300 to 850 °C under a N_2_ stream (600 mL min^−1^) for 30 min.

Carbon-SiO_2_-ZrO_2_ (C-SZ) carbon–ceramic membranes were fabricated by hot-coating the SZ-PB resin sol diluted to 0.25 wt% onto a prefabricated substrate support (Appendix A) preheated to 200 °C followed by pyrolysis at the desired final temperature as shown by the sequence in Appendix A. Two kinds of membranes were fabricated for this work according to the final pyrolysis temperature: C-SZ550 and C-SZ750 with final pyrolysis temperatures of 550 and 750 °C, respectively.

### 2.3. Characterization of SiO_2_-ZrO_2_-Polybenzoxzine-Derived and Carbon-SiO_2_-ZrO_2_ Films, Powders, and Membranes

The presence and transformation of structural moieties in thin films supported on UV-treated Si-wafers were monitored using Fourier Transform-Infrared spectroscopy (FT-IR, FTIR-4100, JASCO, Tokyo, Japan). Water contact angle measurements were carried out on films coated on Si-wafers and measured at room temperature with 0.1 μL drops (Dropmaster DM 300, Kyowa Interface Science Co. Ltd., Saitama, Japan). The decomposition properties of the SZ-PB resin gels and C-SZ powders were analyzed and monitored using thermogravimetry (DTG-60 Shimadzu Co., Kyoto, Japan) and mass spectroscopy-supported thermogravimetry (TG-MS, TGA-DTA-PIMS 410/S, Rigaku, Tokyo, Japan). The presence and the chemical states of constituent atoms were confirmed using X-ray photoelectron spectroscopy (XPS; Shimadzu Co., Kyoto, Japan). The cross-section morphology and the elemental analysis of the carbon–ceramic membrane were examined via Field Emission-Scanning Electron Microscopy (FE-SEM, Hitachi S-4800, Tokyo, Japan). Prior to examination, carefully cut pieces of the membrane were attached to sample holders via carbon tape and vacuum-dried at 50 °C for 24 h. Furthermore, N_2_ sorption of powders was analyzed at −196 °C using BELMAX sorption equipment (Microtrac Bell Co. Ltd., Osaka, Japan). Prior to this measurement, adsorbed gases and vapors were evacuated from the samples at 200 °C for at least 12 h.

### 2.4. Evaluation of Gas Permeance and Hydrothermal Stability

Figure 1 shows a flow diagram of the set-up for evaluating the membrane permeation characteristics and for hydrothermal stability testing. After fabrication, a C-SZ membrane was inserted into the membrane module in the permeation measurement rig at 300 °C under a moderate helium flow of 100 mL min^−1^ for about 12 h to remove the adsorbed vapor and for membrane post-treatment. Single-gas permeation tests of the membranes were carried out using high-purity gases (H_2_, He, CO_2_, N_2_, CH_4_, CF_4_, and SF_6_—in that order). Each gas was fed to the upstream of the membrane module at 200–500 kPa of absolute pressure under temperatures ranging from 50 to 500 °C. Permeate side pressure was kept at atmospheric pressure and the permeate gas flow was measured using a bubble film flow meter (HORIBA-STEC, Horiba Ltd., Kyoto, Japan). The gas permeance (mol m^−2^ s^−1^ Pa^−1^) was then calculated by dividing the measured permeate flow rate by the product of the membrane effective surface area and the transmembrane pressure difference. It should be noted that the range of observed experimental error based on precision for the measured flow rates of gases was controlled to within ±5%.

Hydrothermal stability tests were carried out at 500 °C by feeding the H_2_O + N_2_ mixture to the membrane module such that the partial pressures of steam were 90 and 150 kPa making up 30 and 50 mol% of the mixtures, respectively. Water was pumped from a tank at a steady rate using a liquid chromatography pump and was then vaporized before mixing with N_2_ in a mixer. A stage cut (ratio of the permeate flow rate to the feed flow rate) of lower than 0.167 was maintained. The compositions of the feed, retentate and permeate streams were analyzed by employing the mass balance of the H_2_O-N_2_ binary system. To evaluate the transmembrane pressure-drop of each component, the logarithmic mean pressure difference (Δ*P_i,lm_*) was applied (Equation (5)).
(5)$$\Delta P_{i,lm}=\frac{\Delta P_{i,in}-\Delta P_{i,out}}{\ln\left(\Delta P_{i,in}/\Delta P_{i,out}\right)}$$

In Equation (5), Δ*P_i,in_* and Δ*P_i,out_* represent the difference in partial pressures of component *i* between the retentate side and the permeate side at the inlet and outlet of the module, respectively, that is Δ*P_i,in_* = *P_i_ (feed side inlet)* − *P_i_ (permeate side inlet)* and Δ*P_i,out_* = *P_i_ (feed side outlet)* − *P_i_ (permeate side outlet)*.

## 3. Results and Discussion

### 3.1. Thermal Crosslinking of SiO_2_-ZrO_2_-Polybenzoxazine and Carbon-SiO_2_-ZrO_2_ Formation

In our previous work, the successful formation of a polybenzoxazine-modified SiO_2_-ZrO_2_ was established [25]. The resin of a SiO_2_-ZrO_2_ and polybenzoxazine inorganic-organic hybrid was formed from vinyl trimethoxysilane (VTMS), zirconium n-butoxide (ZrTB), and a benzoxazine compound 3-(3-oxo-1,4-benzoxazin-4-yl) propanoic acid (BZPA) precursors. The polymerization of benzoxazines is commonly accomplished by various methods including organic radical initiation, cationic radical initiation, and thermal polymerization [26]. The SiO_2_-ZrO_2_-polybenzoxazine hybrid was formed by curing at a relatively low temperature of 90 °C, which allowed the opening and propagation of an oxazine ring in the presence of a dibenzoyl peroxide organic radical initiator. However, the curing of benzoxazines is known to improve at temperatures well above 90 °C [27,28]. Hence, in the current work, a curing temperature of 200 °C was applied. Figure 2 shows a schematic representation of the idealized stages in the formation of a highly cured SiO_2_-ZrO_2_-polybenzoxazine. With the help of a radical initiator, the oxazine ring opens to form a tri-substituted benzene ring during stage 1 bonding to phenolic -OH and to a dangling -C-N-C- branch. Subsequent thermal application result in a curing process involving the crosslinking of the opened oxazine and the vinyl groups during stage 2.

This formation model was confirmed using ATR-assisted Fourier Transform Infrared Spectroscopy (FT-IR) to observe any appearance or disappearance of chemical moieties at each formation stage. Figure 3 shows the FTIR spectra of a SiO_2_-ZrO_2_-polybenzoxazine resin before and after thermal curing at 90 and 200 °C. At first, the spectrum of the fresh as-prepared film was measured as a control, as shown in Figure 3 (black line spectrum), and the peaks arising from the different moieties can be observed in the figure. After hydrolysis and condensation reactions, dibenzoyl peroxide (BzO_2_) radical initiator was added to the prepared sol. Peaks representing the dibenzoyl peroxide appeared at 1766 and 1226 cm^−1^. After curing the sol at 90 °C over a 4–5-day period, a gel-like resin formed and the FTIR spectra (red line) revealed that the 90 °C-cured resin showed a spectrum similar to that of the fresh as-prepared sample with the only difference being the disappearance of the BzO_2_ peaks. This indicates that either the curing process did not proceed at 90 °C or it only occurred to a very minimal degree.

Curing at a higher temperature was attempted by directly drying the as-prepared SiO_2_-ZrO_2_-benzoxazine sol at 200 °C. The FTIR profile of the resulting resin shown in Figure 3 (blue line spectrum) reveals the disappearance and appearance of certain peaks. The C=C peak at wavenumber 1603 cm^−1^ was ascribed to the vinyl group of VTMS and was significantly reduced in intensity along with the peak belonging to the BZPA ligand at 1688 cm^−1^. New peaks simultaneously appeared at several points indicating the formation of new moieties: 1223; 1741; 2854; and 2926 cm^−1^ [28]. Both sets of observations may indicate the curing process involving a crosslinking of vinyl and opened oxazine rings since polymerizable moieties belonging to VTMS and BZPA were consumed simultaneously.

The SiO_2_-ZrO_2_-polybenzoxazine resin cured at 200 °C was pyrolyzed to form carbon-SiO_2_-ZrO_2_. The inert calcination of the SiO_2_-ZrO_2_-polybenzoxzine resin resulted in a decomposition of the polymer chain into a sp^2^ type of carbon and the ceramic transformation of the -C-Si-O-Zr- phase to form a carbon-SiO_2_-ZrO_2_ composite [25]. Figure 4a shows the thermogravimetric (TG) profile of SiO_2_-ZrO_2_-polybenzoxzine decomposition under an inert atmosphere. There was an onset of decomposition at 200 °C, which continued with a sharp decline until about 500 °C indicating the breakdown of the organic polymer chain into volatile products. The rate of weight loss beyond 500 °C became slight until 1000 °C was reached. In this range of temperatures, weight loss could be attributed to the process of carbonization whereby carbon was converted into sp^2^ from sp^3^ forms [25,29,30].

The non-volatile residue obtained at 1000 °C after the TG analysis was then subjected to further thermogravimetric analysis under an oxidative atmosphere (He + O_2_ mixture; 20.2% O_2_ by volume) accompanied by mass spectrometry, as shown in Figure 4b. This was performed to quantitatively prove the presence of free carbon in the carbon-SiO_2_-ZrO_2_ obtained after pyrolysis of the SiO_2_-ZrO_2_-polybenzoxzine hybrid. After exposure of the sample to the oxidative gas mixture, no weight loss was observed up to 500 °C. This suggests that below 500 °C the amount of free energy needed to initiate a spontaneous oxidative reaction was insufficient. Therefore, carbon-SiO_2_-ZrO_2_ should be oxidatively stable below 500 °C. However, the sample began to show a loss of weight beyond 500 °C and reached a stable value at around 700 °C. As mass spectrometry showed, the observed peak of *m/z* = 44 that aligned with the weight loss in the sample corresponded to the release of CO_2_ gas. This conclusively proves the presence of free reactive carbon that is released as CO_2_ in a reaction such as C_(s)_ + O_2(g)_ → CO_2(g)_.

### 3.2. The Effect of Pyrolysis Temperature on the Microstructural, Surface, and Membrane Permeation Properties of Carbon-SiO_2_-ZrO_2_

The conversion of carbon from the sp^3^ state to sp^2^ during pyrolysis beyond 500 °C is usually accompanied by microstructural changes such as the rearrangement of graphitic sp^2^ carbon strands into stacked sheets, which results in improved molecular sieving properties [31,32,33]. In the current study, the microstructural change occurring during the pyrolysis of SZ-PB between 550 and 750 °C was investigated despite the small weight loss observed between these two temperatures using nitrogen adsorption–desorption measurements. Figure 5 shows the nitrogen adsorption–desorption isotherms at 77 K of C-SZ powders derived from 200 °C-cured SZ-PB resin pyrolyzed at 550 and 750 °C, where both samples showed type I isotherms characteristic of microporous materials. Compared with the C-SZ750 sample, the C-SZ550 sample adsorbed a higher amount of nitrogen, which indicates that the higher pyrolysis temperature of 750 °C resulted in a densified structure with a smaller surface area and pore width (calculated from NLDFT with pore distribution shown in Appendix A). The shrinking of the pore size with increase in the pyrolysis temperature is a common feature of polymer-derived carbon molecular sieve membranes (CMSMs), and this allows them to finely separate gas species [34,35,36].

Examinations of the surface property differences between C-SZ550 and C-SZ750 revealed possible differences in the chemical states of the free carbon. The interaction of the surfaces with water at room temperature is a good indicator of completeness or otherwise of the pyrolysis transformation of carbon. Figure 6 shows the water adsorption isotherms of C-SZ550 and C-SZ750 powders measured at 25 °C. In carbon materials, the adsorption isotherms typically show negligible amounts of adsorbed water at low relative pressures (p/p_0_ of 0–0.2 typically), but rapid uptakes thereafter [37,38,39]. On the other hand, water adsorption in hydroxylated silica-based materials usually follows a type II isotherm whereby a monolayer adsorption of water occurs at lower p/p_0_, and there is an onset of multilayer adsorption thereafter [40,41,42]. The C-SZ550 and C-SZ750 water adsorption isotherms in Figure 6 therefore seem to show an aggregate of water adsorption isotherms in silica and carbon-based materials, and despite a smaller surface area, C-SZ750 shows a higher amount of water adsorbed compared with that by C-SZ550 (Figure 5). Moreover, C-SZ750 powders show a Langmuir-type isotherm indicating a stronger interaction with water compared with that of C-SZ550. Furthermore, in contrast to silica where hydrophobicity is achieved by removing surface silanol groups by calcination at higher temperatures [42,43], carbon-SiO_2_-ZrO_2_ materials showed an opposite trend.

These observations were backed up by measurements of the water-contact angles of C-SZ550 and C-SZ750 films, as shown in the insets of Figure 6. The C-SZ550 film showed a higher contact angle of 94.8° compared with that of 53.2° for C-SZ750. Based on this result, it appears that C-SZ550 film retains some of the hydrophobic moieties present in the fresh and cured films (Appendix A) that also exhibited high water-contact angles. This shows that the pyrolysis transformation of carbon was not complete at 550 °C. The reduction in the water-contact angle for C-SZ750 indicates the loss of the hydrophobic moieties, which gives rise to free stable carbon. Therefore, the choice of pyrolysis temperature in fabricated carbonized materials is important not only from the perspective of precursor weight loss, but also the chemical state of the non-volatile residue.

Thus, supported C-SZ membranes were fabricated from SZ-PB at 550 and 750 °C. The cross-sectional morphologies of the resulting supported membranes appear in Appendix A, respectively, via FE-SEM. Continuous thin layers of C-SZ can be formed irrespective of the pyrolysis temperature. In addition, carbon-SiO_2_-ZrO_2_ exhibit homogeneous amorphous structures without crystalline phase segregations at both 550 and 750 °C [25]. Therefore, continuous, and uniform separation layers were formed. Figure 7 shows the kinetic diameter dependence of single-gas permeance measured at 300 °C for C-SZ550 and C-SZ750 membranes. Gases with molecular diameters ranging from 0.26 to 0.55 nm were used to probe the molecular sieving properties of the membranes: He (0.26 nm), H_2_ (0.289 nm), CO_2_ (0.33 nm), N_2_ (0.364 nm), CH_4_ (0.38 nm), CF_4_ (0.48 nm), and SF_6_ (0.55 nm). In C-SZ550 membrane, these gases showed an order of permeance (H_2_ > He > CO_2_ > CH_4_ > N_2_ > CF_4_ > SF_6_) that does not conform to the order of their molecular diameters. The anomaly in this order is specific to the higher permeance of H_2_ over He and CH_4_ over N_2_, respectively. This is a result of the loose pore size distribution of the C-SZ550, which allows Knudsen selectivity of H_2_ (2 g mol^−1^) over He (4 g mol^−1^) and that of CH_4_ (16 g mol^−1^) over N_2_ (28 g mol^−1^), respectively, due to the preference of molecular mass over molecular size [44]. As a result, only low H_2_ selectivity of 5, 12, and 11 were achieved over CO_2_, N_2_, and CH_4_, respectively. Nonetheless, the C-SZ550 membrane still showed respectable H_2_/CF_4_ and H_2_/SF_6_ permeance ratios of 210 and 2800, respectively.

On the other hand, the C-SZ750 membrane showed a narrower pore size distribution, which was evident by the sharp order of gas permeance (He > H_2_ > CO_2_ > N_2_ > CH_4_) that followed the order of the molecular diameters. The narrower pore size distribution of C-SZ750 compared with that of the C-SZ550 membrane agrees with observations from the nitrogen-adsorption experiment. The sharp decline in gas permeance from H_2_ to CO_2_ indicates that the C-SZ750 membrane possesses a distribution of pore sizes with the highest probability of falling between the molecular diameters of H_2_ and CO_2_. This can be attributed to the higher fabrication temperature allowing the complete pyrolytic and ceramic transformation of the carbon and -Si-O-Zr- phases to generate a narrower and denser pore structure. The result is high selectivity of H_2_ over CO_2_, N_2_, and CH_4_ of 27,139, and 1026, respectively, with H_2_ permeance (2.8 × 10^−7^ mol m^−2^ s^−1^ Pa^−1^) only slightly lower than that of the C-SZ550 membrane at 5.2 × 10^−7^ mol m^−2^ s^−1^ Pa^−1^ (Appendix A).

Figure 8a,b show the plots of He, H_2_ and N_2_ permeance for the C-SZ550 and C-SZ70 membranes, respectively, as a function of inverse temperature. The average pore sizes of C-SZ550 and C-SZ750 membranes were obtained using the modified-gas translational model (m-GT) proposed by Lee et al. [45], the values of which appear in Table 1 while the pore-size distributions are shown in Appendix A. The values of the apparent activation energies of He, H_2_, and N_2_ permeation were calculated using Equation (6) [46].
(6)$$P_{i\;}=\frac{k_{0,i}}{\sqrt{M_{i}RT}}exp\left(-\frac{E_{p,i}}{RT}\right)$$

In Equation (6), the pre-exponential factor $k_{0,i}$ expresses the combination of configurational factors of the membrane and the permeating molecule, $E_{p,i}$ is the apparent activation energy of permeation for a gas species, *i*, while $M_{i}$ is the molecular weight of the gas species, R is the universal gas constant, and T is the permeation temperature. The obtained values indicate that at 0.56 nm, the average pore size of the C-SZ550 membrane is larger than the molecular diameter of SF_6_. Thus, in Figure 8a, N_2_ shows a Knudsen type of permeation mechanism whereby the gas permeance reduces as permeation temperature increases, giving the apparent activation energy of −1.1 kJ mol^−1^, which is typical of membranes with a loose pore-size distribution. On the other hand, the permeance of both He and H_2_ was increased with temperature showing positive activation energies of 5.0 and 2.6 kJ mol^−1^, respectively (Table 1), which is indicative of activated diffusion.

The permeation of He, H_2_, and N_2_ through the C-SZ750 membrane showed an activated diffusion mechanism. The activation energies of these gases (He: 7.5 kJ mol^−1^, H_2_: 6.5 kJ mol^−1^, N_2_: 13.5 kJ mol^−1^) are much higher than in the C-SZ550 membrane, which supports calculations for the pore size of the C-SZ750 membrane at 0.4 nm, because a higher amount of energy is required to permeate the smaller pores of the C-SZ750 membrane. It should also be noted that the activation energy of N_2_ was much higher than those of He and H_2_ because gases with larger molecular diameters require higher amounts of energy to permeate smaller pores.

### 3.3. Hydrothermal Stability of Carbon-SiO_2_-ZrO_2_ Membranes

Having examined the different permeation properties of C-SZ550 and C-SZ750 membranes, hydrothermal stability tests were carried out to investigate the roles that the different microstructural and surface properties of both membranes played in their durability under steamed conditions. Figure 9 shows the time courses at 500 °C for He and N_2_ permeance under dry and hydrothermal conditions of 90 and 150 kPa of steam partial pressure (30 and 50 mol% steam, respectively) for the C-SZ550 membrane. After steady states of He and N_2_ permeance (5.4 × 10^−7^ and 2.1 × 10^−8^ mol m^−2^ s^−1^ Pa^−1^, respectively) were established at 500 °C, a mixture composed of 30 mol% of steam and 70 mol% of N_2_ was fed to the membrane until steady-state fluxes of both N_2_ and trapped H_2_O were achieved. At a steam partial pressure of 90 kPa, the steady value of N_2_ permeance was significantly lower than that for a dry state. This reduction in N_2_ permeance could have been due to the blocking effect of water molecules as both water and N_2_ molecules permeated the micropores. After 90 kPa of steam treatment for 23 h and complete drying of the membrane, the steady-state N_2_ permeance of 3.8 × 10^−8^ mol m^−2^ s^−1^ Pa^−1^ was higher than the starting permeance of 2.1 × 10^−8^ mol m^−2^ s^−1^ Pa^−1^ before hydrothermal treatment, while He permeance was only marginally increased. Therefore, an enlargement of the pores had occurred during the hydrothermal treatment. Further hydrothermal testing at 150 kPa for 9 h revealed a similar trend of reduced N_2_ permeance under hydrothermal conditions. However, after stable N_2_ and H_2_O permeance were achieved and the membrane was returned to the dry state, He and N_2_ permeance were permanently reduced to 3.8 × 10^−7^ and 9.8 × 10^−9^ mol m^−2^ s^−1^ Pa^−1^, respectively, from the initial dry state values. This indicates a densification of the C-SZ structure upon exposure to more severe hydrothermal conditions. Thus, it is evident that the C-SZ550 membrane was unstable during long-term hydrothermal testing.

The procedure carried out for the C-SZ550 membrane was repeated for the C-SZ750 membrane. Figure 10 shows the time course at 500 °C for He and N_2_ permeance under dry conditions and for N_2_ permeance under hydrothermal conditions for the C-SZ750 membrane. As with the C-SZ550 membrane, in the C-SZ750, N_2_ also showed a reduced permeance in the presence of steam compared with that of the dry-state. Following the removal of steam and drying for several hours, the permeance values for He and N_2_ (5.2 × 10^−7^ and 6.6 × 10^−9^ mol m^−2^ s^−1^ Pa^−1^, respectively) were unchanged compared with that before hydrothermal stability testing. After further exposure at 150 kPa steam partial pressure, the permeance values for He and N_2_ (5.7 × 10^−7^ and 7.5 × 10^−9^ mol m^−2^ s^−1^ Pa^−1^, respectively) were only slightly increased after drying and reaching a confirmed steady state. This shows a substantially more stable performance under hydrothermal conditions for the carbon-SiO_2_-ZrO_2_ membrane fabricated at 750 °C.

Figure 11a,b show the plots of single-gas permeance at 300 °C as a function of kinetic diameter for C-SZ550 and C-SZ750 membranes, respectively, before and after the hydrothermal stability tests under steam with partial pressures of 90 and 150 kPa. As noted from the discussion on the time course of hydrothermal stability for the C-SZ550 membrane, the permeance of N_2_ increased after the hydrothermal stability test at 90 kPa steam partial pressure. Figure 11a shows that the permeance values for CH_4_, CF_4_, and SF_6_ also increased significantly after the hydrothermal test, but the membrane continued to retain a molecular sieving property. The selectivity of H_2_ over N_2_ and CH_4_ was reduced from 21 to 17 and 35 to 19, respectively (Appendix A). This suggests the formation of a looser pore size distribution, which allowed the permeance of large gases to increase. The permeance values for He and H_2_ remained largely the same as they were permeating the smaller micropores yet unaffected by the steam permeation. After the hydrothermal stability test at 150 kPa partial pressure of steam, the pore size distribution was significantly changed. The permeance values for He, H_2_, CO_2_, N_2_, and CH_4_ were reduced quite considerably while those for CF_4_ and SF_6_ were increased. The ratios of the permeance of H_2_ over those of N_2_ and CH_4_ were both increased to 23, due to the densification of the micropores (Table 1). However, the ratios of N_2_ and CH_4_ permeance values to those of CF_4_ and SF_6_ then approached Knudsen selectivity. This type of property is characteristic of a bimodal pore size distribution and suggests a loss of molecular sieving ability by the membrane.

In Figure 11b, the kinetic diameter dependence of single-gas permeance in the C-SZ750 membrane before and after hydrothermal stability tests tell a different story. An ideal membrane for application in alcohol steam reforming reactions to produce CH_4_-free H_2_ must be able to maintain a H_2_/CH_4_ selectivity equal to or greater than 100 [12]. The C-SZ750 membrane not only maintained the integrity of the pore structure, but also maintained a high H_2_ permeance and H_2_/CH_4_ selectivity of 3.8 × 10^−7^ mol m^−2^ s^−1^ Pa^−1^ and 439, respectively, after exposure to hydrothermal conditions of 150 kPa of steam partial pressure at 500 °C.

The observations during the hydrothermal stability tests of C-SZ550 and C-SZ750 membranes were confirmed via X-ray photoelectron spectroscopic (XPS) examinations of C-SZ550 and C-SZ750 films supported on Si wafers before and after exposure to steam. Figure 12a,b show the narrow C 1s XPS spectra of C-SZ550 and C-SZ750 films, respectively, before and after steam treatment conditions of 90 kPa steam partial pressure at 500 °C for 3 h. In Figure 12a, the observed C 1s spectra (black line) of C-SZ550 before and after steam treatment can be deconvoluted into four constituent peaks each representing the status of carbon bonds. Before steam treatment, the more prominent peak at 60.5% represents the existence of sp^3^ hybridized carbon with only 5.5% of sp^2^ hybridized carbon formed. This supports the idea that pyrolytic transformation was yet to be complete at 550 °C. Following steam exposure, the proportion of sp^3^ to sp^2^ carbon changed considerably. Sp^3^ carbon then made up 33.2% while sp^2^ carbon made up 30.3%. It could thus be concluded that steam had a deleterious effect on the carbon atoms still bonded in the sp^3^ state.

In Figure 12b, sp^2^ carbon was the more prominent peak for C-SZ750 before steam treatment with a proportion of 58.8% compared with 29.4% for sp^3^ carbon. This shows a more complete pyrolytic transformation of SiO_2_-ZrO_2_-polybenzoxazine at 750 °C. After the exposure to steam, C-SZ750 retained similar proportions of sp^2^ and sp^3^ carbon at 52.5 and 20.8%, respectively, compared with those before steam treatment. This, therefore, indicates that C-SZ750 is more stable than C-SZ550 in the presence of steam. The prior conversion of more sp^3^-type carbon to sp^2^ at a higher pyrolysis temperature provided greater resistance to adverse hydrothermal conditions.

Figure 13a,b present a schematic summary of the observations on the hydrothermal stability performances of C-SZ550 and C-SZ750 membranes, respectively. XPS examination has shown that there is a large proportion of sp^3^ hybridized carbon in C-SZ550, as discussed above. A detailed discussion on the effects of pyrolysis temperature on the pyrolysis mechanism and the nature of residual carbon can be found in a previous work [25]. These sp^3^ carbons may represent incompletely pyrolyzed carbon chains due to low pyrolysis temperature. From the observations in Figure 9 for the C-SZ550 membrane, it appears that the initial exposure to steam at 90 kPa partial pressure caused a gasification of incompletely carbonized organic moieties leaving larger pores and exposing the decarbonized -Si-O-Zr- linkages. This phenomenon was also reported by Duke et al. [18]. A more severe steam exposure at 150 kPa partial pressure then resulted in hydrolysis of the exposed -Si-O-Zr- linkages forming -Si-OH and -Zr-OH groups and to migration and subsequent recondensing under dry conditions [16], as illustrated in Figure 13a. This recondensation leads to a densification of the C-SZ550 membrane and a resultant reduction in permeance. For a C-SZ750 membrane, Figure 13b illustrates a structure with a complete ceramic and carbonized form. Upon exposure to steam atmosphere, free sp^2^ carbon, being stable, shields most of the -Si-O-Zr- linkages from H_2_O attack. Some outlying -Si-O-Zr- bonds could be hydrolyzed but the resulting -Si-OH and -Zr-OH groups are prevented from migration and thus recondensation by the free carbon, as suggested by Duke et al. [18].

Yoshida et al. [47] investigated the hydrothermal stability of several unmodified SiO_2_-ZrO_2_ membranes. While a SiO_2_-ZrO_2_ membrane with a silica-to-zirconia ratio of 9/1 showed the best balance of hydrothermal stability and H_2_/N_2_ selectivity at 500 °C and 100 kPa partial pressure of steam, the pore size distribution of the membrane changed considerably as hydrothermal exposure increased. The selectivity of H_2_ over N_2_ increased but the permeance of all gases reduced considerably and the membrane assumed a bimodal pore size distribution. Ahn et al. [48] also tested the hydrothermal stability of an unmodified SiO_2_-ZrO_2_ membrane (10% ZrO_2_) at 650 °C under a steam partial pressure of 101 kPa. After the steady state of the membrane was established at the testing conditions, H_2_ permeance was found to have reduced by 56%, although with a corresponding increase in H_2_ selectivity over N_2_. These two studies have shown that a trend of reduced H_2_ permeance with a corresponding selectivity increase is to be expected after pure SiO_2_-ZrO_2_ membranes are exposed to high-temperature hydrothermal conditions. However, a sustained high permeance of H_2_ is important in membrane reactor applications. This work has thus shown that the modification of SiO_2_-ZrO_2_ via carbonization can be utilized in achieving hydrothermally robust membranes with sustained levels of H_2_ permeance and selectivity.

## 4. Conclusions

Carbon-SiO_2_-ZrO_2_ was formed from the pyrolysis of polybenzoxazine-modified SiO_2_-ZrO_2_ under an inert atmosphere. The chosen pyrolysis temperature determined the surface and microstructural properties of the resulting carbon-SiO_2_-ZrO_2_ composite. Pyrolysis of the cured SiO_2_-ZrO_2_-polybezoxazine membrane at 750 °C resulted in a sharp pore size distribution compared with pyrolysis at 550 °C, which also resulted in higher H_2_ selectivity over CO_2_, N_2_, and CH_4_. After conducting hydrothermal stability tests at 500 °C under steam partial pressures of 90 and 150 kPa, a carbon-SiO_2_-ZrO_2_ membrane fabricated at 750 °C showed stable performance under the hydrothermal conditions applied with sustained levels of H_2_ permeance and H_2_ selectivity attributed to the complete pyrolytic formation of free carbon which prevented structural changes in the ceramic membrane backbone during hydrothermal exposure.

## Figures and Tables

**Figure 1 membranes-13-00030-f001:**
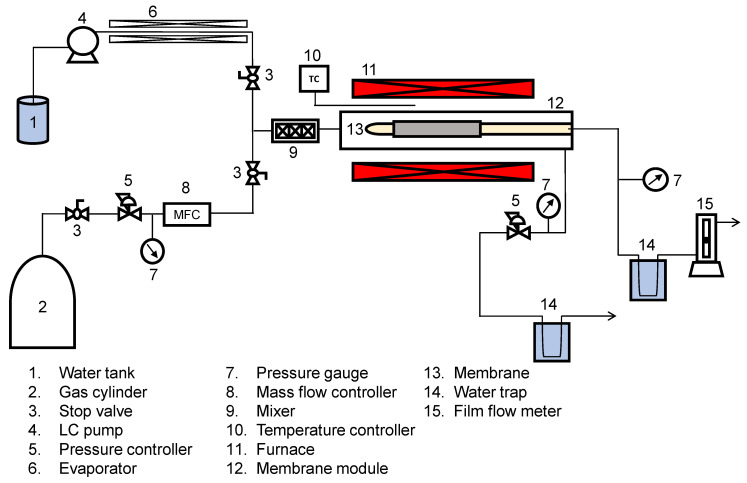
Schematic flow diagram of the membrane evaluation set-up.

**Figure 2 membranes-13-00030-f002:**
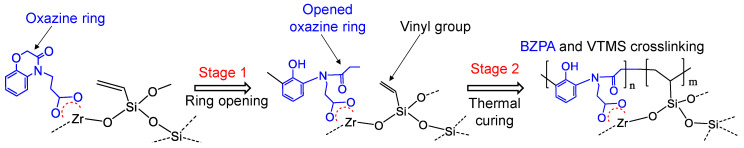
Schematic representation of the thermal crosslinking of the vinyl group of VTMS and of the oxazine ring of BZPA in a SiO_2_-ZrO_2_-polybenzoxzine hybrid.

**Figure 3 membranes-13-00030-f003:**
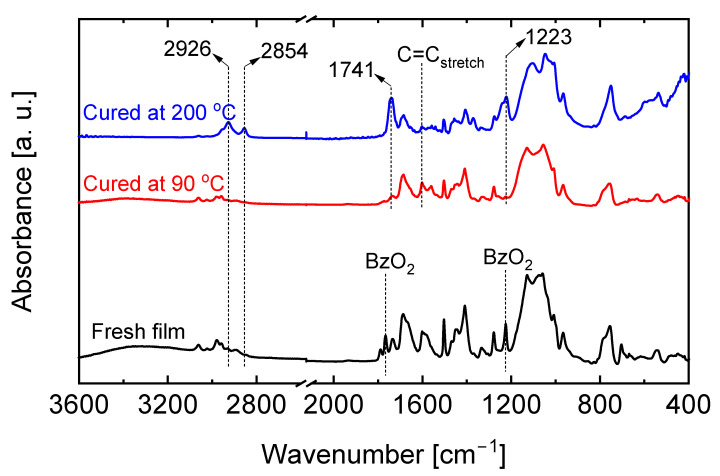
FTIR spectra of SiO_2_-ZrO_2_-polybenzoxzine before and after thermal curing at 90 (red line) and 200 °C (blue line).

**Figure 4 membranes-13-00030-f004:**
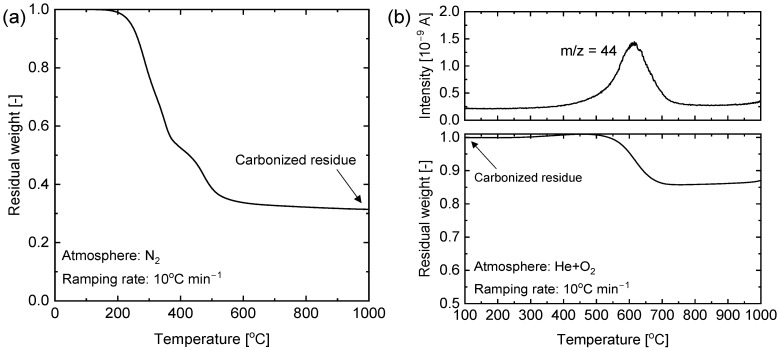
(**a**) Thermogravimetric analysis (TG) of the decomposition under a N_2_ atmosphere of SiO_2_-ZrO_2_-polybenzoxzine resin cured at 200 °C. (**b**) Thermogravimetric analysis and mass spectrometry (TG-MS) of the decomposition under an oxidative atmosphere of the non-volatile residue derived following the TG procedure in (**a**).

**Figure 5 membranes-13-00030-f005:**
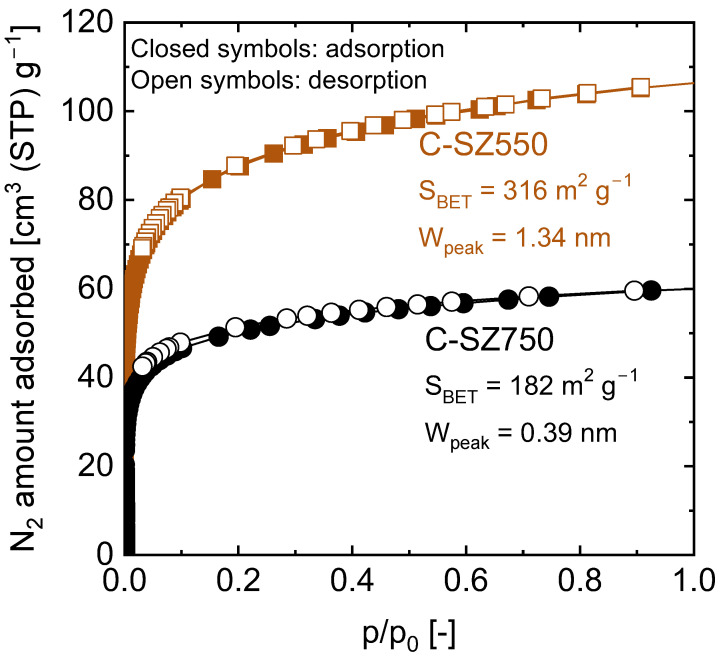
Nitrogen adsorption and desorption isotherms at 77 K of carbon-SiO_2_-ZrO_2_ (C-SZ) powders derived from 200 °C-cured SiO_2_-ZrO_2_-polybenzoxzine resins after pyrolysis at 550 and 750 °C (S_BET_: Bruneuer–Emmett–Teller surface area; W_peak_: peak pore width).

**Figure 6 membranes-13-00030-f006:**
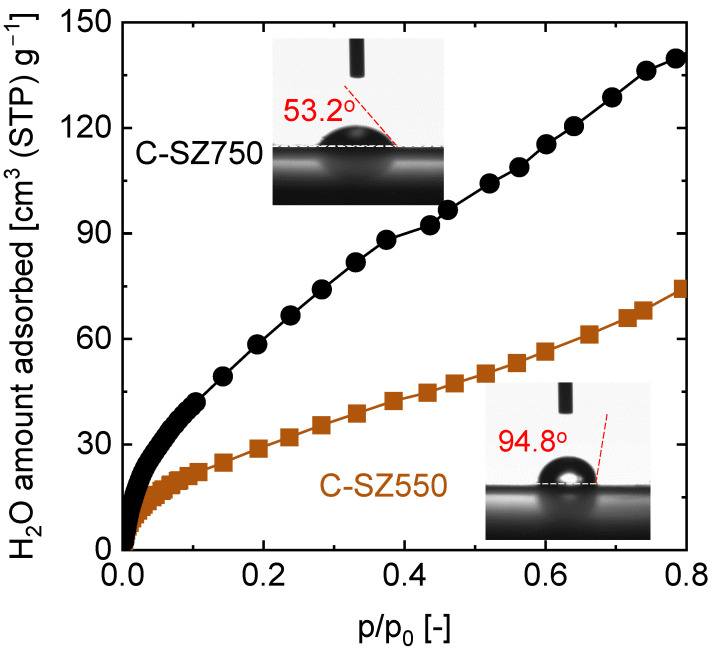
H_2_O adsorption isotherms measured at 25 °C of carbon-SiO_2_-ZrO_2_ (C-SZ) powders derived after the pyrolysis of 200 °C-cured SiO_2_-ZrO_2_-polybenzoxzine resins at 550 and 750 °C (Inset: Water-contact angles of respective carbon-SiO_2_-ZrO_2_ films).

**Figure 7 membranes-13-00030-f007:**
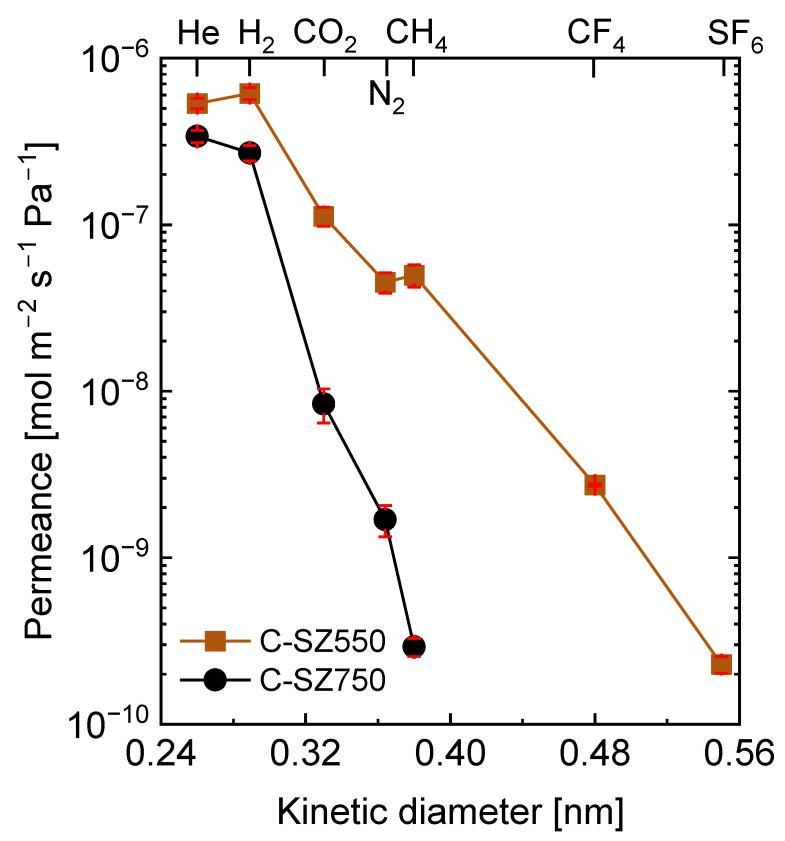
Single-gas permeance at 300 °C as a function of kinetic diameter for carbon-SiO_2_-ZrO_2_ membranes prepared at 550 and 750 °C.

**Figure 8 membranes-13-00030-f008:**
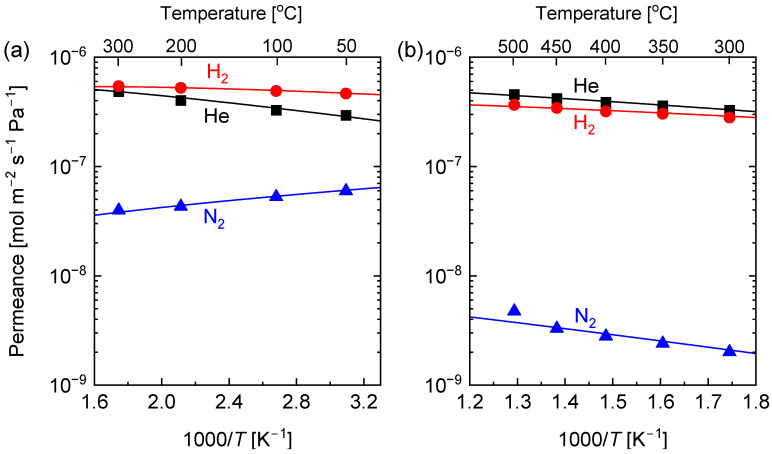
Temperature dependence of the He, H_2_ and N_2_ permeance of carbon-SiO_2_-ZrO_2_ membranes prepared at (**a**) 550 and (**b**) 750 °C.

**Figure 9 membranes-13-00030-f009:**
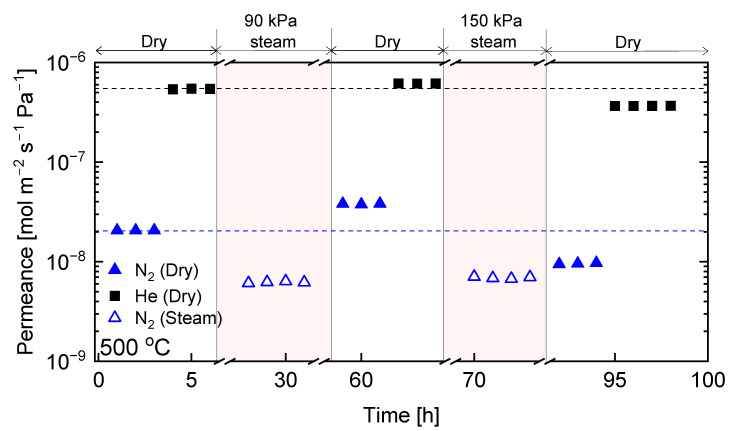
Time courses of hydrothermal stability tests at 500 °C in a mixture of H_2_O + N_2_ (90 and 150 kPa of steam partial pressure; total pressure of 300 kPa) for carbon-SiO_2_-ZrO_2_ membranes prepared at 550 °C.

**Figure 10 membranes-13-00030-f010:**
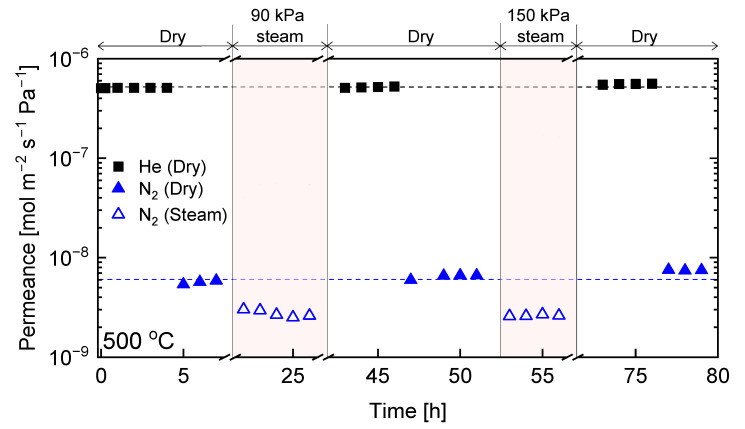
Time courses for hydrothermal stability tests at 500 °C in a mixture of H_2_O + N_2_ (90 and 150 kPa of steam partial pressure; total pressure of 300 kPa) for carbon-SiO_2_-ZrO_2_ membranes prepared at 750 °C.

**Figure 11 membranes-13-00030-f011:**
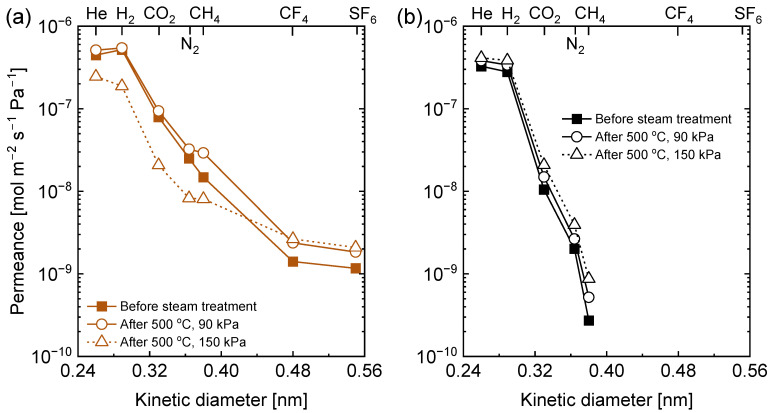
Single-gas permeance as a function of kinetic diameter measured at 300 °C of carbon-SiO_2_-ZrO_2_ membranes fabricated at (**a**) 550 and (**b**) 750 °C before and after hydrothermal stability tests at 500 °C with steam partial pressures of 90 and 150 kPa.

**Figure 12 membranes-13-00030-f012:**
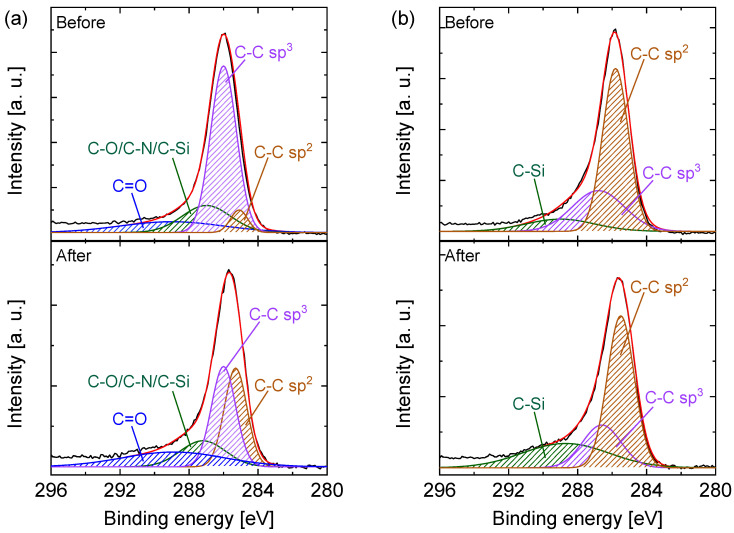
C 1s X-ray photoelectron spectroscopy of carbon-SiO_2_-ZrO_2_ films prepared at (**a**) 550 and (**b**) 750 °C before and after 3 h of steam treatment at 500 °C, 90 kPa steam partial pressure.

**Figure 13 membranes-13-00030-f013:**
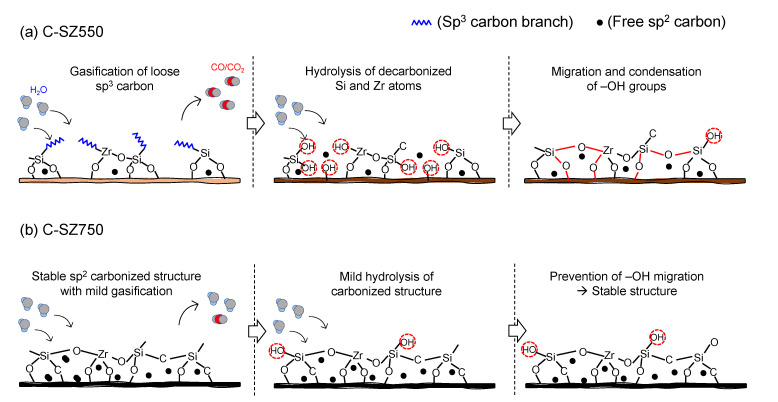
Schematic illustration of the role of fabrication temperature in the microstructural stability of carbon-SiO_2_-ZrO_2_ membranes fabricated at (**a**) 550 and (**b**) 750 °C upon exposure to steam.

**Table 1 membranes-13-00030-t001:** Carbon-SiO_2_-ZrO_2_ membrane properties determined from the temperature dependence of He, H_2_, and N_2_ permeance values.

Fabrication Temperature [°C]	Average Pore Size ^a^ [nm]	Activation Energies [kJ mol^−1^]		
		He	H_2_	N_2_
550	0.56	5.0	2.6	−1.1
750	0.40	7.5	6.5	13.5

^a^ Determined by the modified gas-translational model.

## Data Availability

The data presented in this study are available on request from the corresponding author.

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
