# Peer review of "Hydrothermal Stability of Hydrogen-Selective Carbon–Ceramic Membranes Derived from Polybenzoxazine-Modified Silica–Zirconia"

_membranes, 2022, doi:10.3390/membranes13010030_

Round 1
Reviewer 1 Report
The manuscript discusses hydrothermal stability of hydrogen-selective carbon-ceramic membranes derived from polybenzoxazine-modified silica-zirconia (carbon-SiO2-ZrO2 membranes). The significance of content is high and the subject is important to development of hydrogen separation technology, which is crucial to the hydrogen economy. It is a promising area worthy of investigation. The text is generally well-written and commendable efforts can be observed in the preparation of this manuscript. However, there is room to strengthen the manuscript further and revision would be necessary to address the following comments:
1)
Under Abstract (Lines 21-22)
“This work thus demonstrates the promise of carbon SiO2-ZrO2 membranes for H2 production…”
It may be more precise to replace the wording ‘H2 production’ with ‘H2 separation’.
2)
Under Section 1 (Lines 38-43).
Is dehydrogenation of propane the intended application for the carbon-SiO2-ZrO2 membranes?
Please also discuss and elaborate on other application fields for separation of H2.
3)
Under Section 1 (Lines 45-47).
Instead of just listing the traditional separation techniques, provide a more detailed literature review and a summary table to describe and compare the different separation techniques (pressure swing adsorption, cryogenic distillation, membrane separation, etc.) in terms of their principles, energy consumption, purity of H2 produced, advantages and disadvantages, current state of development, etc.
4)
Section 2.2 (Line 152)
This is a lab-scale study with a relatively low flow of 100 mL/min.
Discuss what is the range of experimental error, and what are the plans to scale up the study to demonstration and commercial scale.
5)
Section 2.2 (Line 168-169)
The manuscript mentioned: “The compositions of the feed, retentate and permeate streams were analysed by employing the mass balance of the H2O-N2 binary system...”
How about analyses of the other gases such as H2, CH4, CO2, etc…?
Please discuss and elaborate.
6)
Figure 7 and Figure 8.
Were the experiments repeated to produce statistically significant result? Please also include experimental error bar or standard deviation in the figures.
7)
Figure 9 and Figure 10.
For ease of reference, please provide legends for the different data symbols in the figures.
8)
Section 4 Conclusions.
It was written that the fabrication temperature of the carbon-SiO2-ZrO2 membranes affects the separation performance (Permeance and H2 selectivity etc.). For clarity of presentation, provide a graph or table to discuss on the observed relations.
9)
Section 4 Conclusions.
Under this section, the manuscript may add on with a discussion on the challenges and future prospective of the said carbon-SiO2-ZrO2 technology and how exactly it contributes towards the roadmap for further development of H2-economy.
Reviewer 2 Report
General comment: This is an interesting study which reported the long-term hydrothermal performance of composite carbon-SiO2-ZrO2 membranes and demonstrated its promise for H2 production under severe hydrothermal conditions. The manuscript is recommended to be published after addressing the following issues:
1. In your previous work J. Mater. Chem. A, 2020, the novel preparation of carbon-SiO2-ZrO2 membrane has already been reported. Therefore, the significance of its utilization in H2 production under severe hydrothermal conditions should be emphasized more to highlight the contribution of current work as the innovation point.
2. What is the reason for selecting 550 ℃ and 750 ℃ to investigate the microstructural change occurring during the pyrolysis of SZ-PB? Since pyrolysis at 750 ℃ performance better than 550 ℃, why not choosing one more higher pyrolysis temperature as a further comparison? It is not very convincing to explore the effect of pyrolysis temperature on properties of carbon-SiO2-ZrO2 with only two examples.
3. A more precisely directed explanation is suggested to highly generalize the effect of pyrolysis temperature, and a clearer direction is expected to be pointed on optimizing the material preparation in further application.
4. How is the uniformity of the prepared carbon-SiO2-ZrO2 membrane? There seems no relevant exploration in the manuscript.
5. Please check the section numbers carefully. There is more than one repetition of “2.2” and “3.1” in the manuscript which may cause reading confusion.
6. In the Abstract, more concise expressions should be used to summarize the main achievements of the article and highlight the novelty of this work to immediately arouse interests of the readers. Some detailed data can be appropriately omitted. The same suggestion also applies in the Conclusion section.
7. Only about one-fourth of the cited literatures are within last five years. More recent literatures are recommended to the follow the latest research advances.
